

# A protocol for the intercomparison of marine fishery and ecosystem models: Fish-MIP v1.0

Derek P. Tittensor[1,2], Tyler D. Eddy[2,3], Heike K. Lotze[2], Eric D. Galbraith[4,5], William Cheung[3], Manuel Barange[6,7], Julia L. Blanchard[8], Laurent Bopp[9], Andrea Bryndum-Buchholz[2], Matthias Büchner[10], Catherine Bulman[11], David A. Carozza[12], Villy Christensen[13], Marta Coll[14,15], John P. Dunne[16], Jose A. Fernandes[7,17], Elizabeth A. Fulton[11,18], Alistair J. Hobday[11,18], Veronika Huber[10], Simon Jennings[19,20,21], Miranda Jones[3], Patrick Lehodey[22], Jason S. Link[23], Steve Mackinson[19], Olivier Maury[24,25], Susa Niiranen[26], Ricardo Oliveros-Ramos[27], Tilla Roy[9,28], Jacob Schewe[10], Yunne-Jai Shin[25,29], Charles A. Stock[16], Philip J. Underwood[1], Jan Volkholz[10], James R. Watson[26], Nicola D. Walker[19]

1 United Nations Environment Programme World Conservation Monitoring Centre, 219 Huntingdon Road, Cambridge, CB3 0DL, U.K.
2 Department of Biology, Life Sciences Centre, Dalhousie University, 1355 Oxford Street, Halifax, N.S., B3H 4R2, Canada
3 Nippon Foundation-Nereus Program, Institute for the Oceans and Fisheries, The University of British Columbia, Vancouver, B.C., V6T 1Z4, Canada
4 Institució Catalana de Recerca i Estudis Avançats (ICREA), 08010 Barcelona, Spain
5 Institut de Ciència i Tecnologia Ambientals (ICTA) and Department of Mathematics, Universitat Autonoma de Barcelona, 08193 Barcelona, Spain
6 Fisheries and Aquaculture Policy and Resources Division, Food and Agriculture Organisation of the United Nations (FAO), Rome, 00153, Italy
7 Plymouth Marine Laboratory, Prospect Place, The Hoe, Plymouth, PL13 DH, U.K.
8 Institute for Marine and Antarctic Studies, University of Tasmania, 20 Castray Esplanade, Battery Point. TAS. 7004, Private Bag 129, Hobart, TAS 7001, Australia
9 Institut Pierre-Simon Laplace / Laboratoire des Sciences du Climat et de l'Environnement, CNRS / CEA / UVSQ, CE Saclay, Orme des Merisiers, F-91191 Gif sur Yvette, France
10 Potsdam Institute for Climate Impact Research, Telegrafenberg A56, 14473 Potsdam, Germany
11 CSIRO Oceans and Atmosphere, GPO 1538, Hobart, Tasmania 7001, Australia
12 Department of Earth and Planetary Sciences, McGill University, 3450 University Street, Montreal, H3A 0E8, Canada
13 Institute for the Oceans and Fisheries, University of British Columbia, 2202 Main Mall, Vancouver BC, Canada V6T 1Z4
14 Institute of Marine Science (ICM-CSIC), Passeig Marítim de la Barceloneta, nº 37-49, 08003, Barcelona, Spain.
15 Institut de Recherche pour le Développment, UMR MARBEC & LMI ICEMASA, University of Cape Town, Private Bag X3, Rondebosch, Cape Town 7701, South Africa
16 Geophysical Fluid Dynamics Laboratory, National Oceanic and Atmospheric Administration, Princeton University, Princeton, NJ 08540, U.S.A.
17 AZTI, Herrera Kaia, Portualdea z/g, Pasaia (Gipuzkoa), 20110, Spain
18 Centre for Marine Socioecology, University of Tasmania, 20 Castray Esplanade, Battery Point, Tasmania, 7004, Australia
19 Centre for Environment, Fisheries and Aquaculture Science (CEFAS), Lowestoft Laboratory, Lowestoft, NR33 0HT, U.K.
20 School of Environmental Sciences, University of East Anglia, Norwich Research Park, Norwich, NR4 7TJ, U.K.
21 International Council for the Exploration of the Sea, H. C. Andersens Blvd 46, 1553 København V, Denmark
22 CLS, 11 rue Hermes 31520 Ramonville Saint Agne, France
23 National Oceanic and Atmospheric Administration, National Marine Fisheries Service, 166 Water Street, Woods Hole, MA 02543, U.S.A.
24 IRD (Institut de Recherche pour le Développement) - UMR 248 MARBEC, Av Jean Monnet CS 30171, 34203 Sète cedex, France



[25] University of Cape Town, Dept. of Oceanography - International Lab. ICEMASA Private Bag X3, Rondebosch 7701, Cape Town, South Africa

[26] Stockholm Resilience Centre, Stockholm University, Kräftriket 2B, SE-114 19 Stockholm, Sweden.

[27] Instituto del Mar del Perú (IMARPE). Gamarra y General Valle s/n Chucuito, Callao, Perú

[28] ECOCEANA (Ecosystem, Climate and Ocean Analysis), 57 Rue Archereau, Paris, 75019, France

[29] University of Cape Town, Marine Research (MA-RE) Institute, Department of Biological Sciences, Private Bag X3, Rondebosch 7701, South Africa.

*Correspondence to:* Derek Tittensor (derek.tittensor@unep-wcmc.org)

**Abstract.** Model intercomparison studies in the climate and earth sciences communities have been crucial to build credibility and coherence for future projections. They have quantified variability among models, spurred model development, contrasted within- and among-model uncertainty, assessed model fits to historical data, and provided ensemble projections of future change under specified scenarios. Given the speed and magnitude of anthropogenic change in the marine environment, and consequent effects on food security, biodiversity, marine industries and society, the time is ripe for similar comparisons among models of fisheries and marine ecosystems. Here, we describe the Fisheries and Marine Ecosystem Model Intercomparison Project protocol version 1.0 (Fish-MIP v1.0), part of the Inter-Sectoral Impact Model Intercomparison Project (ISIMIP), a cross-sectoral network of climate impact modellers. Given the complexity of the marine ecosystem, this class of models has substantial heterogeneity of purpose, scope, theoretical underpinning, processes considered, parameterizations, resolution (grain size) and spatial extent. This heterogeneity reflects the lack of a unified understanding of the marine ecosystem, and implies that the assemblage of all models is more likely to include a greater number of relevant processes than is any single model. The current Fish-MIP protocol is designed to allow these heterogeneous models to be forced with common Earth System Model (ESM) CMIP5 outputs under prescribed scenarios for historic (from 1950s) and future (to 2100) time periods; it will be adapted to CMIP6 in future iterations. It also describes a standardized set of outputs for each participating Fish-MIP model to produce. This enables the broad characterization of differences between, and uncertainties within, models and projections when assessing climate and fisheries impacts on marine ecosystems and the services they provide. The systematic generation, collation and comparison of results from Fish-MIP will inform understanding of the range of plausible changes in marine ecosystems, and improve our capacity to define and convey strengths and weaknesses of model-based advice on future states of marine ecosystems and fisheries. Ultimately, Fish-MIP represents a step towards bringing together the marine ecosystem modelling community to produce consistent ensemble medium- and long-term projections of marine ecosystems.

# 1 Introduction

The ocean provides nearly half of global primary production (Field et al., 1998), hosts 25% of eukaryotic species (Mora et al., 2011), provides 11% of global animal protein consumed by humans (FAO, 2014),  and is a source of livelihoods for



millions (Sumaila et al., 2012). Yet the pace and magnitude of projected climate change over the coming century, in combination with fisheries exploitation and a raft of other human impacts, suggests that marine ecosystems will remain under considerable pressure in the mid- to long-term (Pörtner et al., 2014; UN, 2016). Identification of the potential future effects of these pressures, even with high uncertainty (Payne et al., 2016), is required to anticipate the impacts of

environmental change on ecosystem resilience (Bernhardt and Leslie, 2013), biodiversity conservation (Cheung et al., 2016a; Queirós et al., 2016), socio-economics (Fernandes et al., 2017) and food security (Barange et al., 2014; Merino et al., 2012). Marine ecosystem models give us an approach to meeting this goal by providing scenario-driven projections of future fisheries production (e.g. Blanchard et al., 2012; Fernandes et al., 2015, 2017; Lehodey et al., 2015; Mullon et al., 2016), marine ecosystem structure and functioning (Jennings and Collingridge, 2015) and species compositions and distributions

(Jones and Cheung, 2015) under global change.

The scientific understanding of the physical climate system and its response to anthropogenic perturbation has profited enormously from model intercomparison efforts like the Coupled Model Intercomparison Project (CMIP) (Taylor et al., 2012) and the Ocean Model Intercomparison Project (OMIP) (Griffies et al., 2016). CMIP and other efforts have highlighted differences among models, provided ranges of potential climate change responses and ensemble projections for end-users,

and allowed the outputs of individual analyses to be interpreted in a broader context. They have also provided a quantification of the relative contributions of different sources of uncertainty to projected uncertainties in climate responses (Hawkins and Sutton, 2009; Payne et al., 2016).

In addition to model intercomparison experiments for the climate and ocean system, a systematic intercomparison and assessment of *impact* models – including the marine realm – is similarly essential for understanding the impacts of (and

associated uncertainty around) climate change on important biological and human systems (Barange et al., 2014). Such impact models typically use the outputs of scenario-driven earth system models (ESMs), individually or as ensembles, as inputs to project the effects of these on sectors such as agriculture or energy. The Inter-Sectoral Impact Model Intercomparison Project (ISIMIP; www.isi-mip.org) was set up to enhance consistency among climate impact studies across different sectors, including food production, ecosystems and biodiversity, freshwater availability, and human health among

others (Huber et al., 2014; Schellnhuber et al., 2013). It does so chiefly through providing common climate and socio-economic input data and defining a common set of simulation experiments (Warszawski et al., 2013).

Although there have been prior intercomparisons of fisheries and marine ecosystem models they have been limited to a few models applied to local or regional case studies (Coll et al., 2008; Fulton and Smith, 2004; Jones et al., 2013; Shin et al., 2004; Smith et al., 2011; Travers et al., 2010), or to lower trophic levels of the global-ocean rather than the whole ecosystem

[see the Marine Ecosystem Model Intercomparison Project MAREMIP, pft.ees.hokudai.ac.jp/maremip/index.shtml; (Bopp et al., 2013)]. Here, we describe the Fisheries and Marine Ecosystem Model Intercomparison Project (Fish-MIP) protocol v1.0, which intends to standardize to the extent possible input variables to fisheries and marine ecosystem models, and to analyze, compare and disseminate outputs from multiple models to assess climate and fisheries impacts on marine ecosystems and the services they provide, such as potential future fisheries catches. The Fish-MIP protocol has been designed in coordination



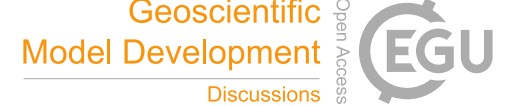

with the other ISI-MIP sectors and forcings, which will enhance consistency when looking at synergistic impacts and considering multi-sectoral aspects such as global food security and economic impacts. This article describes the Fish-MIP protocol and provides background on the range of existing ecosystem models, including the resolved processes, theoretical approaches, and specifications, and provides the foundation for forthcoming studies within the umbrella of this project. We

emphasize here that Fish-MIP is continuously ongoing in terms of development and refinement: see www.isimip.org/gettingstarted/marine-ecosystems-fisheries/ for updates. While many Fish-MIP model runs for the v1.0 protocol (ISI-MIP simulation round 2A) have been completed, some are still in progress or under analysis, and results will be published once runs are complete (though see Blanchard et al., 2017).

The scope of Fish-MIP v1.0 is global and regional using fisheries and marine ecosystem models able to make historical

(~1950s onwards) and medium to long-term (defined here as ~2030-2100) projections of ecosystem structure, dynamics and function using the same set of scenarios and reanalyses of climate change and variability. Single- and multi-species tactical models for fisheries stock assessment and management are therefore excluded; a complementary initiative presently underway under the auspices of the intergovernmental organizations ICES (www.ices.dk) and PICES (www.pices.int), the Strategic Initiative on Climate Change effects of Marine Ecosystems (SICCME) aims to fill this gap. The long time-horizon

of Fish-MIP means that outputs are most likely to inform those focusing on long-term changes in the global and regional environment and future policy development, such as the Intergovernmental Panel on Climate Change (IPCC), the Intergovernmental Platform on Biodiversity and Ecosystem Services (IPBES), and the United Nations Regular Process of work on the Sustainable Development Goals (SDGs). Results may also be of interest to national bodies and management authorities interested in scenarios or species distribution shifts (ICES, 2016). Questions of interest include the effect of

climate impacts on the distribution, diversity and productivity of fishes and fisheries, exploration of global fisheries scenarios and projections and their implications for food security (Béné et al., 2015), and the conservation status of marine fauna and their role in biogeochemical cycles.

The development of marine ecosystem models, which aim to simulate the structure, dynamics, production, and functional role of marine biota interacting with each other and their environment across multiple trophic levels, was initiated at least

four decades ago (Andersen and Ursin, 1977; Polovina, 1984; Sheldon et al., 1977), and somewhat earlier for efforts with less resolved biology (Hamblin, 2005). More recently, to support ecosystem-based fisheries management, marine ecosystem models have been applied to assess changes in ecosystem structure and function under fisheries and environmental drivers (Fulton, 2010). Unlike in the physical and chemical sciences, where there is often clarity about fundamental representations and processes driven by underlying theory or experimentation, the development of marine ecosystem models has been

approached from many perspectives, reflecting differences in scientific and management objectives, theoretical frameworks, modelling structures and parameterizations, input data needs, resolutions (spatial, temporal, vertical, process, and taxonomic), and process complexity. They include differing assumptions of top-down, bottom-up or mixed trophic control, the role of species as opposed to trophic groups, functional groups, or body-size classes, and the characterization of growth, mortality, recruitment and movement.



Applying such a range of different model types will provide useful insights into the effects of climate variability and change on marine ecosystems and fisheries, but also makes intercomparisons challenging. Unsurprisingly, the inputs to and outputs from such models are diverse and difficult to standardize, with no common set of defined output metrics that can be used for comparison purposes. Here we review the models participating in the first round of Fish-MIP simulations (using

CMIP5 ESM output), describe our intercomparison protocol, highlight the challenges that have arisen in developing this protocol, and detail the approaches that we have used to resolve these difficulties. We also identify future pathways for Fish-MIP, including the use of CMIP6 output and refined fisheries scenarios. The lessons learned here also apply to other marine model comparisons, and will help to guide the development of new models to investigate patterns of change in the future oceans.

**2. Marine ecosystem models participating in Fish-MIP**

The last three decades have seen a profusion of marine ecosystem models being developed, with many emerging during the last decade (Fulton and Link, 2014; Nielsen et al., 2017; Peck et al., 2016). Fish-MIP is open to all developers and users of marine ecosystem models who are willing to run consistent scenarios to facilitate comparisons. All models in the intercomparison must be documented in appropriate venues, such as the peer-reviewed literature, to ensure that descriptions

of the model are widely available, and that key features and parameterizations are codified and model runs repeatable. Here we introduce the model types that have been included in Fish-MIP to date, recognizing that these are a subset of the many available and extensively reviewed elsewhere (Fulton and Link, 2014; Hollowed et al., 2000; Plagányi, 2007; Plagányi et al., 2011, 2014; Townsend et al., 2008; Travers et al., 2007).

**2.1 Model heterogeneity**

The diversity of model types participating in Fish-MIP, and some of their unique characteristics, are summarized in Table 1. One constant feature across all the participating models is the inclusion of multiple species or functional groups (typically an aggregation of species or food web elements) and environmental drivers. These minimum specifications are in line with the need to characterize the transfer of biomass from primary producers to mid- and upper trophic level organisms, which are often those impacted or used by society or of conservation interest. Among the key differences therein, the spatial resolution

of participating models ranges from simple 0D boxes encompassing model parameters averaged over a large area; to irregularly shaped polygons corresponding to depth and bathymetric features; to regional models gridded at 0.1×0.1 degrees; to global models gridded at resolutions typically around 0.5×0.5 to 1×1 degrees. Some models are fully or partially vertically resolved, while others consider depth implicitly through food web interactions and habitat preference patterns, or do not model the vertical dimension at all. The movement of fish can be ignored, defined by discrete rules between adjacent grid

cells, driven by climate niche models, or expressed through formal advection-diffusion systems of equations. Fisheries are





elaborated to differing degrees, from complete absence, to a simple fishing mortality term, to more elaborate fishing effort allocation formulations.

The wide array of modeling approaches leads to a wide array of input data requirements (Table 2). Some models use a single forcing variable, often a primary production anomaly or estimate derived from the output of a regional ocean-
biogeochemistry or a global earth-system model (ESM), while others use multiple variables directly. While all the global Fish-MIP models use temperature as an input variable, some use fully vertically resolved 3D data, others use temperature averaged over a near-surface layer such as the mixed-layer; a few wide layers (e.g. epi-, upper meso- and lower mesopelagic layers); sea-surface and sea-bottom temperature; or sea-surface temperature only (Table 2). Some models require or can use additional inputs other than primary production and temperature, such as total alkalinity, nutrients, light (photosynthetically-
active radiation) or a turbulent mixing parameter. In terms of temporal scales, most models run on monthly or yearly time-steps, though others run on very fine time scales of a day or less, making for another axis of variation when considering model differences and input requirements.

In terms of model outputs, all current Fish-MIP models can produce a measure of biomass density for all consumers and for particular size classes (which itself may require translation from functional groups to sizes classes, or classification of
species under a particular class if body length is not tracked) (Table 3). Most models also consider some measure of fisheries production (e.g. catches, fisheries landings, mortality rates). It is worth highlighting the difference between models that predict catch from effort and fish biomass versus those that are forced using catch data: in the former, catch is an output, while in the latter, it is an input, and measures of fisheries production cannot thus be calculated. Further, some models enable statistical fitting to catch or effort data in which case other parameters are estimated (such as fishing mortality rates).

The great heterogeneity in input data requirements presented a large challenge when developing the Fish-MIP protocol, and constrained the set of ESM outputs (from CMIP v5) that could be used. Forcing using identical ESM outputs is not feasible as the requirements and options for each Fish-MIP model differ (Table 2). Instead, participating models are forced using standardized inputs for those variables that are included (Table 3). These variables are used on the spatial, temporal, and vertical scale appropriate to each model, but with a minimum monthly time-step – e.g., models with a daily time-step
had to use monthly forcing data, taking the average value for that month and applying it daily (without the day-to-day variability typical of finer scale forcing). This primarily reflects the limitations imposed by the available ESM output.

Across the diversity of participating models, we recognize four broad classes: those focusing on species distribution, trophodynamic structure, size- or age-based structure, and composite (hybrid) models. This simple classification is used to structure this summary of models contributing to the Fish-MIP project.

**2.1.1 Model classes: species-distribution based models**

Species distribution models (Cheung et al., 2016b; Fernandes et al., 2013; Jones and Cheung, 2015; Pearson and Dawson, 2003) use statistical, empirical and theoretical relationships between a species and its environment to explore the implications of shifting environmental conditions and resulting habitat suitability distributions on the biomass and spatial



range of species. Recent development has integrated this class of models with mechanistic representation of ecophysiology and population dynamics and hence potential fisheries production (Cheung et al., 2011; Fernandes et al., 2013), and fishing scenarios (Fernandes et al., 2016, 2017). For instance, the Dynamic Bioclimate Envelope Model (DBEM) (Table 1) was applied to a suite of over 1,000 species to examine shifts in distribution, abundance, productivity under climate change

scenarios and resultant global patterns of local extinction, invasion, biodiversity and catch (Cheung et al., 2016b; Jones and Cheung, 2015). A version of DBEM that has incorporated size-based trophodynamics to mimic ecological interactions has also been applied to model a number of ecosystems (Fernandes et al., 2013, 2016, 2017). Typically, this class of models include a large number of primarily commercially valuable fishes and invertebrates.

### 2.1.2 Model classes: trophodynamic based models

Trophodynamic models are typically structured based on species interactions and the transfer of energy across trophic levels. Ecopath with Ecosim (EwE), one of the oldest and most widely used marine ecosystem modelling approaches (Christensen and Walters, 2004), focuses explicitly on trophodynamics, as does its global offshoot EcoOcean (Christensen et al., 2015). EwE has been extensively used to explore potential fisheries impacts on and management options for aquatic ecosystems, to assess the impact of other human activities and climate variability and change (Niiranen et al., 2013), and to analyze and

compare ecosystem structural and functional traits (Colleter et al., 2015). More recent applications include cumulative human impacts, marine *conservation*, environmental impact assessments and end-to-end modelling (Coll et al., 2015). EwE models typically include demersal and pelagic species from primary producers up to top predators, both commercial and non-commercial. Ecospace, the spatial-temporal model run in Ecospace in conjunction with the food web and fisheries dynamics components, has been further developed to be able to spatially derive the foraging capacity of individual species

from physical, oceanographic, and environmental drivers such as depth, temperature bottom type, oxygen concentrations and primary production (Christensen et al., 2014). This development, in combination with the recently added spatial-temporal framework module (Steenbeek et al., 2013), has bridged the gap between environmental envelope models and food web models (Christensen et al., 2014, 2015). EwE models are typically structured by species or functional groups, and can also include age- or size- based representation of species.

### 2.1.3 Model classes: size- or age- based models

Contemporary models in this class build on size-based conceptualizations of marine ecosystems (Boudreau et al., 1991; Dickie et al., 1987; Platt and Denman, 1978; Sheldon et al., 1972; Sheldon and Parsons, 1967) to characterize the flux of energy from primary producers to higher predators. Size-based approaches are predicated on the substantial role of body size in structuring food webs, which results from the dominance of small primary producers, size-based predation and

ontogenetic increases in trophic level when many predators grow 5–6 orders of magnitude in body mass from egg to adult (Jennings et al., 2012). The size-based models with the lowest parameter demands rely on empirical relationships that link body mass, temperature and biological rates to support parameterization (e.g. Benoît and Rochet, 2004; Blanchard et al.,



2009; Borgmann, 1987; Jennings and Collingridge, 2015; Watson et al., 2015). More complex size-based approaches describe some of the differences among species in a size-structured community by incorporating information on traits such as species' maximum (asymptotic) size (e.g., Andersen and Beyer, 2006; Carozza et al., 2016; Pope et al., 2006). Life history theory can be used to estimate parameters such as size at maturity and reproductive output from maximum size. Other size-

and species- based models may incorporate some species-specific information directly (Blanchard et al., 2014; Maury et al., 2007; Maury and Poggiale, 2013; Shin and Cury, 2004), but use general size-based relationships to describe other components of the system such as predator-prey relationships. As a variant of a size-based approach, time of development (i.e. age) to reach a critical life stage can be used to model the dynamics of species or functional groups (Lehodey et al., 2010). This approach can help to represent the effects of key environmental influences (e.g., temperature) in a different way.

As well as being used for assessing the effects of fishing and environmental variation on marine ecosystems (e.g., Fu et al., 2013; Shin and Cury, 2004), size- or age- based ecosystem models have been used to underpin linked analyses of the effects of climate change on fisheries and society (Barange et al., 2014), including marine commodity trade (Mullon et al., 2016).

### 2.1.4 Model classes: composite (hybrid) models

The final class of models use multiple formulation types to create representations of entire systems. Until recently, these

models were distinct from other classes due to the breadth of processes covered; convergent evolution means that this is becoming less the case. Nevertheless, this class of models still tend to feature a broader set of ecological processes (including movement, feeding, reproduction, habitat use), the major biophysical drivers (e.g. temperature and salinity), a more complete food-web, and often nutrient dynamics and cycles. This is achieved through composite (hybrid) end-to-end approaches that bring together multiple modelling methods either by coupling component models (which may be from the

classes above) or via direct integration in a single unified framework. They resolve the food web to at least functional group and in some cases species level (or a mix of the two approaches). An example is Atlantis (Fulton et al., 2011), which uses a transport model to characterize three dimensional current flows, a size resolved biomass pool-based representation of the plankton food web, patch dynamic representation of demersal habitats, colonization-based representation of bacterial groups and a fully-age structured representation of the vertebrate groups. On top of this is an effort allocation and fisheries

management model, and an ability to capture biogeochemical processes.

Many of the composite models (e.g. OSMOSE; Travers et al., 2009) are age- or size-structured allowing them to capture the size-based feeding and ontogenetic shifts found in the size-based approaches discussed above. Aspects of the size-based approach are also used to represent the bulk of the food-web in some composite models such as for the mid-trophic levels of SEAPODYM (Lehodey et al., 2008, 2010) and APECOSM (Maury, 2010), with more detailed elaboration applied to a sub-

set of the system of particular interest (e.g. target species and higher trophic level species, such as tunas). Most composite (hybrid) process based models have had a regional focus, in part due to data and computational requirements. An exception is the Madingley model (Harfoot et al., 2014), which although not designed specifically for fisheries (or indeed only marine) studies, can be applied to those questions nonetheless. It takes an agent- and process-based approach and through



representing broad functional groups rather than individual species has somewhat mitigated computational constraints. As with the other classes of models there has been a broad range of motivations for the development of these type of models. Nevertheless, their most common uses to date have been to explore ecosystem dynamics (Harfoot et al., 2014), consider fisheries management options (Fulton et al., 2014), climate change scenarios (Lehodey et al., 2015) and test the performance

of ecosystem indicators (Fulton et al., 2005; Travers et al., 2006).

## 3. Forcing data: Earth-system models and fisheries

The fundamental goal of Fish-MIP is to compare the response of marine ecosystem models to common external forcings, including anthropogenic change. This is achieved by forcing the fisheries and marine ecosystem models with ocean

hindcasts and scenario-driven projections from General Circulation Models (GCMs) that include coupled biogeochemistry modules. At present, this has been limited to global-scale ESMs following CMIP5 protocols, though in principle regional GCMs (e.g. using the ROMS framework) could also be used. For participating Fish-MIP models that allow it, the simulations also include spatially-explicit estimates of fishing effort or catch. A completely standardized forcing cannot be used as the broad range of ecosystem models require different sets of inputs. Excluding all models except those with

common inputs would have removed many well-established and widely used marine ecosystem models from the Fish-MIP project and substantially reduced the inclusivity and utility of the comparisons. Consequently, Fish-MIP decided to force all models with the specific inputs they needed, but to draw these from a consistent set of ESM simulations (subsampled at different spatial and temporal resolutions).

The first round of Fish-MIP was conducted with CMIP5 output, as detailed here. This is because model outputs from

CMIP6 in formats suitable for marine ecosystem models were not available to the ISI-MIP and Fish-MIP communities at the time the Fish-MIP v1.0 simulations were started. However, the Fish-MIP project has participated in the Vulnerability, Impacts, Adaptation and Climate Services (VIACS) advisory board for CMIP6 (Ruane et al., 2016), specifically to communicate the requirements for marine ecosystem models, and in particular the need for archiving of full three-dimensional depth-resolved monthly biogeochemical outputs. We therefore anticipate that CMIP6 output (Eyring et al.,

2016) will be utilized in a future round of Fish-MIP (likely in ISI-MIP phase 3, planned for end of 2018), and in addition that the number of ESMs providing suitable forcings to Fish-MIP models will increase.

### 3.1 Environmental drivers from Earth-system models

We reviewed the 10 CMIP5 models considered by Bopp et al. (2013) that included projections for a suite of potential physical and biogeochemical stressors (warming, deoxygenation, acidification, and changes in ocean productivity).  The

models we determined suitable for Fish-MIP were GDFL-ESM2M and IPSL-CM5A-LR, with CESM1-BGC also likely to be incorporated in the future. Four key criteria were used to select ESM model outputs for use in Fish-MIP:



*(i)*     *Availability*: At minimum, representative concentration pathway (RCP) simulations with lowest and highest impact scenarios (RCPs 2.6 and 8.5) to 2099 are available and accessible, and ideally RCPs 4.5 and 6.0 as well. Historical runs are available from at least 1960. All physical and biogeochemical oceanic forcing fields needed to drive all the Fish-MIP marine ecosystem models (i.e. all 'common' variables listed in Table 2) are, ideally, available at 3D spatial and monthly-mean temporal resolution.

*(ii)*     *Quality control*: ESMs vary widely in the complexity of their biogeochemical formulations, with some including minimalist representations aimed at efficient representation of carbon cycling and others featuring more resolved representations of plankton dynamics aimed at both biogeochemical and marine resource applications. While the ESMs are robustly correlated with SST, oxygen, $CO_2$ and 3D resolved pH, they vary widely in their correlation to satellite based NPP estimates (Bopp et al., 2013), likely in large part a result of differences in the scope of objectives. Globally, we did not select models where the correlation coefficient ($r$) with NPP fell below 0.4.

*(iii)*     *Future response*: The selected ESM models spanned a significant fraction of the cross-ESM range in the future projections of physical and biogeochemical fields, especially in primary production and plankton biomasses. This allows the Fish-MIP models to be tested across a wide range of plausible future scenarios. In this regard, IPSL-CM5A-LR features a relatively strong surface warming and global NPP decline, GFDL-ESM2M has relatively small changes, and CESM1-BGC, though not incorporated in the first round of model runs, is an intermediate case.

*(iv)*     *Model drift:* The model drift of the ESM outputs, as diagnosed from the control simulation (i.e. no climate-forcing), is negligible.

Interestingly, the 'availability' criterion imposed the greatest limitation on the choice of ESMs. Of the more than 30 ESMs participating in CMIP5, only a subset (10) included necessary marine biogeochemical ocean model components (temperature, pH, dissolved oxygen, and NPP) (Bopp et al., 2013). Furthermore, only one model (IPSL-CM5A-LR) currently produces the full set of ESM outputs required to drive all marine ecosystem models included in the Fish-MIP project (i.e., met the full 'availability' criterion and had full three-dimensional depth-resolved monthly data). Many modelling groups at the time of protocol development had either not uploaded their full biogeochemical fields to the CMIP5 Earth System Grid Federation (ESGF) archive or did not output the variables at the full 3D spatial and monthly time resolution required by some Fish-MIP marine ecosystem models, possibly due to a lack of time and/or funding. As a result, ESM modelling groups were approached individually to obtain access to the full biogeochemical fields at the required temporal and spatial resolution. 'Quality control' also eliminated several models from our ESM selection. Since ESMs have not been specifically designed to force ecosystem models, some outputs such as planktonic biomass and productivity may have spurious or unrealistic values for some regions and/or depths, and needed to be checked before use. Although 'availability' and 'quality control' mostly acted to limit our ESM selection, fortuitously the models selected for these





attributes also span a range of potential 'future response' trajectories in ocean temperature, NPP, dissolved oxygen and pH (Bopp et al., 2013). For ease of use, all data were converted into the same units and re-gridded onto a common 1x1 degree grid. For examples of ESM forcing data see Figures S1 and S2.

The number and type of phytoplankton and zooplankton groups represented in ESMs vary substantially (Bopp et al.,
2013) and an explicit differentiation into large and small planktonic groups, as needed by some of the ecosystem models, was not always available. For the purpose of forcing Fish-MIP models, 'large' and 'small' phytoplankton groups were defined in such instances. In the ESM outputs, we defined the large phytoplankton functional group (*lphy*) to include diatoms, large non-diatoms and the diazotrophs; although small diazotrophs exist, it was generally not possible to separate them out, and ESMs tend to parameterize diazotrophs as larger, trichodesmium like organisms (Capone et al., 1997). Primary
production associated with nitrogen fixation is much less than total primary production (Gruber and Galloway, 2008), allaying concerns over this simplification. The small phytoplankton group (*sphy*) included the pico- and nanophytoplankton groups. Only the IPSL-CM5A model explicitly represented size-differentiated zooplankton groups (*szoo* and *lzoo*). For other ESMs where these were unavailable, the zooplankton size-classes were post-diagnosed by normalizing to phytoplankton biomass such that: lzoo = zoo×lphy/(sphy+lphy) and szoo = zoo×sphy/(sphy+lphy). This simple approach makes the
assumption that the small zooplankton and large zooplankton biomass residence times are the same.

For the first round of Fish-MIP, all modellers were encouraged to force their models with the ESM inputs that made sense biologically and ecologically, as determined by the mechanisms and assumptions specific to individual models (the 'optimized' simulation), enabling us to examine outputs based on ideal (from the perspective of the individual marine ecosystem model) forcings. For subsequent rounds, we will also specify a 'standardized' ESM input simulation to better
distinguish differences in marine ecosystem model outputs due to ESM forcings from those due to ecosystem model structure. As an example, some modeling groups preferred to remove the diazotroph contribution from their primary productivity or planktonic biomass input fields because it was assumed that this material is not efficiently transferred up the food chain. Removing the diazotrophs is the 'optimized' simulation, while the 'standardized' simulation would include the diazotrophs in the biomass and primary productivity input fields.

### 3.1.1 Downscaling to Regional Domains

Fish-MIP aims to compare temporal and spatial outputs between global and regional models. Output from global models can be subsampled over the areas considered by regional models, such as the North Sea, and the regional models can be forced with ESM output averaged over the grid cells included in the regional model domain. In this way, the global and regional model responses to common environmental variations can be directly compared. Regional marine ecosystem models often
have highly resolved survey, stock assessment and/or fishing effort data as inputs, and therefore this direct comparison can help to test and contextualize biases in the global models. Furthermore, regional marine ecosystem models have often been previously integrated with higher resolution ocean and biogeochemical simulations, providing comparisons with resolution of fine-scale structure in space and time.



At this point in ESM development, it is important to note that regional downscaling remains problematic. The ESMs in CMIP5 have ocean resolutions of ~1-2 degrees and are thus only capable of resolving circulation features on the order of 300 km or larger. This leads to limited representation of coastal ocean and marginal sea currents and upwelling and, in some cases, substantial regional biases in ecosystem drivers (Holt et al., 2016; Stock et al., 2011). In addition, ESMs struggle to

represent iron limitation well (Tagliabue et al., 2015), which can add an additional potential source of bias, especially when simulating primary production in iron-limited ecosystems such as the Southern Ocean (Moore et al., 2013) or sub-Arctic Pacific. They also struggle to represent the extent of tropical oxygen minimum zones (Cabré et al., 2015), which represents a limitation for marine ecosystem models using dissolved $O_2$ as an input variable. More generally, confidence in climate change projections is greatest at continental scales and above (Randall et al., 2007).  Thus, while the Fish-MIP protocol is

developed to enable consideration of both global and regional applications, the limitations of present tools suggest an emphasis on forecasted large-scale changes (e.g., shifting and evolving ocean biomes, latitudinal contrasts), coupled with more cautious consideration of the regional implications of large-scale drivers (such as $CO_2$) resolved by ESMs.

To date, some regional ecosystem models have used downscaling of global-scale model outputs or high-resolution shelf seas models (Barange et al., 2014; Stock et al., 2011). Concerted development of high-resolution global climate and earth

system models with improved resolution of coastal processes (e.g., Saba *et al.*, 2016) should ease this limitation moving forward.  Alternatively, growing suites of regionally down-scaled solutions (e.g., Holt *et al.*, 2016) may provide a basis for region-specific implementations of the Fish-MIP protocol. Although it is likely that these biases will be reduced over the coming years and decades, they must be borne in mind as inescapable shortcomings of the current state-of-the-art in ESM models.

## 3.2 Fishing scenarios

Fishing is an important human driver of changes in marine ecosystems and is represented in most marine ecosystem models as a spatially and temporally varying term that removes biomass and production from the system. This term is typically applied in one of two ways within marine ecosystem models. It can be imposed as a biomass removal rate per unit time, based on empirical or modelled catches (or landings), and removed directly from the system biomass for specific functional

groups, ages, and size classes. Alternatively, it can be applied as a mortality rate, which removes a fraction of the existing biomass per unit time. This mortality rate can be applied directly, or calculated from a fishing effort term which considers both 'nominal effort' (total resources devoted to fishing) and the catchability of fish to give an 'effective' fishing effort (Jennings et al., 2001). Nominal effort reflects human involvement in fishing (e.g., number and engine power of fishing boats, and time spent fishing). Catchability, defined as the proportion of biomass that can be caught per unit of fishing effort,

can be affected by both ecological and human factors. For example, the aggregation behavior of some fish stocks can increase their catchability (Arreguin-Sanchez, 1996). Improvements in fishing technology or changes in gear configuration (e.g., changes in the mesh size of nets) can also affect catchability.



There are important differences between models forced with catch and those forced with effort that make the consistent representation of fisheries impacts in the Fish-MIP project challenging when bringing such disparate approaches together. Models forced with catch can drive ecosystem components extinct if the forced catches are incompatible with biomass dynamics, and can be used to see if observed historical catches can be maintained given model dynamics. Models forced

with effort or mortality produce emergent catches derived from available modelled biomass. These catches can be compared to historically observed data to assess confidence in the model's forcing and ability to reproduce observations. Fish-MIP models also vary in the complexity and degree of linkages and feedbacks to other biophysical and human components of the ecosystems that they represent. In some models, fishing effort, catch, or mortality rates are parameters or forcing variables with no feedback from the biological systems to the socio-economic systems. However, some hybrid and trophodynamic

models fully couple these systems.

Individual Fish-MIP models use various data sources for their fisheries impact forcing, which reflects variation in model purpose, development history, functional group representation, spatial scale and resolution, inclusion of illegal, unregulated, and unreported (IUU) fisheries, and other factors. Modelled fishing mortality rates, catches, or effort can vary over time and space and can be spatially explicit or applied at local, regional or basin-scales. Most databases on global fishing catch or

effort can be spatially disaggregated to match the scales represented in (global) marine ecosystem models. The difference between such data sources may be considerable; see Figure S3 for an example of the difference in global catches over time between two databases.

As examples of approaches taken by individual Fish-MIP models, the EcoOcean model applies fishing as an effort term based on the Sea Around Us Project (SAUP) effort database (Anticamara et al., 2011; Christensen et al., 2015; Watson et al.,

2013); the DBEM model uses an alternate catch reconstruction database (Watson, 2017); BOATS has a dynamic bioeconomic approach using SAUP catch price data to simulate spatially-resolved changes in fishing effort over time, based on individual fishers attempting to optimize their outcomes (Carozza et al., 2017); SS-DBEM represents maximum sustainable yield without explicitly calculating fishing mortality (Fernandes et al., 2016, 2017; Mullon et al., 2016), using SAUP data supplemented by other sources (the ICES data collections (www.ices.dk/marine-data/dataset-

collections/Pages/Fish-catch-and-stock-assessment.aspx) and the RAM legacy stock assessment database (www.ramlegacy.org)). This last example further demonstrates how fishing impacts may be modelled implictly. Regional models typically used yet different data sources, often finely-resolved observer-derived local datasets with highly taxonomically resolved information.

In the first round of Fish-MIP, to maximize the participation of marine ecosystem models, we decided to allow models to

implement fishing according to their own standard method, so as to produce realistic historical transients, followed by constant fishing impacts for future scenarios. That is, models continued to use their specific catch, effort, or other forcing data (often but not always SAUP data for global models), but with simple standardized scenarios imposed.

For historical simulations (to 2005), biomass removal based on reported catches or historical effort levels was imposed to reconstruct the historical level of fishing (Christensen et al. 2015), or assumptions were made about average fishing



mortality in historical periods (Cheung et al., 2016b). As above, this was based on each Fish-MIP model's specified fishing database. In addition to the standardized scenario, an optional scenario was suggested with no fishing (zero effort/mortality). Some models included no representation of fishing and thus only ran this optional scenario of zero fishing. For future projection simulations, in the current absence of operationalized spatially explicit scenarios of effort or catch, the

standardized model run was to keep fishing constant at 2005 rates (from their specified data source), while the optional scenario was again zero fishing. Our scenarios therefore imposed a fishing impact (in fished model runs) while maintaining consistency among each model's fit to historical data. Given the simplicity of this approach, we focus on climate impacts for Fish-MIP 1.0, and plan to explore the impacts of fishing in more detail following the development of specified future fisheries scenarios.

Improving the inter-model consistency of historical and future fisheries forcings remains a substantial challenge for the marine ecosystem model inter-comparison community. In subsequent iterations of Fish-MIP, we will aim to reconcile the various historical input data streams to further quantify their effects on results and, if possible, further standardize prescribed input sources. Furthermore, formally-developed global scenarios representing projections of future fishing activity, management, and technological change are beginning to become available (e.g. Maury et al., 2017). When these become

operationalized (made spatially-explicit in a format that is suitable for input into marine ecosystem models), such scenarios will likely be used to replace the standardized run with constant 2005 catch or effort in future scenario model experiments.

## 4. Output data

The broad range of marine ecosystem and fisheries models (Section 2) leads to an equally broad range of potential model

outputs (Table 4). To compare outputs from Fish-MIP models, we selected six common output variables that most models would be able to produce (total system biomass, total consumer biomass, biomass of consumers >10 cm and >30 cm, and, for models forced by fishing effort, catches and landings; see Table 5). We also developed a list of additional optional outputs that some models are capable of producing (e.g., biomass and catch of individual species/trophic groups); see Table S1. This dual approach was adopted to achieve a balance between having common outputs that all models could produce,

and producing comparisons across as broad an array of outputs as possible. Some output variables were not native to individual models (such as biomass by size classes), and required post-processing; however, this appeared to be the only way to compare outputs of size-based models with those from species/trophic group structured models.

Some models output biological state variables (e.g. biomass) as wet weight, and some as carbon, and so different factors were used when converting from one to the other, sometimes differing between functional groups. Another issue is the

conversion of biomass density to or from size-classes, functional groups, and species. There appeared to be no universal approach that was meaningful across all ecosystem model types, so we settled on model outputs for all sizes, as well as size bins of maximum length of >10cm and >30cm. This meant that mass-length conversions were handled differently in different models. All of these details matter when we are seeking to develop effective and informative comparisons, and



were not all readily documented or accessible at the outset. However, our comments should not be taken to imply that every detail needs to be harmonized among models; in fact, for equitable comparison, it may be desirable to retain much of the diversity of model specifications. In any case, carefully specified information must be shared on those aspects that are harmonized – be it variable names and definitions, scenarios, or datasets – and those aspects that necessarily remain

idiosyncratic. For example, 3D models can use oceanic depth profiles that differ from ESM outputs. Ultimately, modelers used their specific depth profile and we accepted this as a potential confounding factor (in the same way as having 2D and 3D models). Another important contrast is the inclusivity of the species represented, which in some cases is all species (e.g. Macroecological), in others all commercial species (e.g. BOATS), and in others only a subset of commercial species (e.g. SEAPODYM). Thus, at this point, for quantities such as fish catch the relative trends can be compared readily between

models and observations, but the absolute values need to be considered carefully.

It was agreed that output data should be column-integrated, on a 1×1 degree grid, and at a monthly (where possible) resolution, with 'no data' values set to 1.0e+20f and variable names as in Table 5. Time-series requested were 1971-2004/2005 (depending on ESM-forcing) for historical models runs, and 2006-2099 for future scenarios. For an example of model output, see Figure S4. All files were to be saved in netcdf format with a .nc4 extension (a conversion script for .csv

files was made available at: http://vre2.dkrz.de). Full details on outputs, including the conventions for file naming were made available in the ISIMIP 2A simulation protocol at https://www.isimip.org/protocol/#isimip2a, and the instructions at https://www.isimip.org/protocol/isimip2b-files/ (also see the Supplementary Material for this paper).

Commercial species were defined as all potentially-harvested fish >10cm. All modelling groups used their own size classes and functional groups when running the model, and provided the name and definition of size classes and functional

groups used for total catch and landings. For common standard equations and mass-length conversion, models that did not have their own conversions were referred to FishBase (www.fishbase.org). It was also requested that the conversion values from wet weight to Carbon should be specified.

## 5. Core simulations in the Fish-MIP v1.0 protocol

As the first marine sector included within ISIMIP, the Fish-MIP protocol was developed to align with the aims and scope of

the overall ISIMIP project, and to harmonize forcing simulations and scenarios whenever feasible. However, we also needed to balance this approach with the need to allow the effective intercomparison of marine ecosystem models given currently available modelling platforms and forcing data (see previous Section), and to consider the critical role of fishing. For historical (hindcasting) model runs, we used a GFDL-reanalysis product (Cheung et al., 2013) as our common 'observational' climate input set of time-series (Table 6). For future projections, we used GFDL and IPSL products (see

Section 3), with priority on the RCP 2.6 and 8.5 scenarios, to span a range of alternate futures (Table 6). Thus, it should be possible to compare the effects of a climate signal based on a given ESM across ecosystem models, with differences among



models reflecting a combination of their sensitivities to different aspects of climate change linked to differences in model structure and parameterization (as well as any other differences in forcing variables).

Depending on ecosystem model complexity, times required for a defined simulation can differ by orders of magnitude. We therefore decided on a multi-tier hierarchy of standardized and optional climate- and fisheries-forced simulations (see Table 6) that ensured we could: (i) compare many marine ecosystem model outputs across a few top priority ESMs and reanalysis product historical runs; (ii) assess the spread of future projections by comparing outputs for at least one ESM (IPSL) across all four RCPs (2.6, 4.5, 6.0, 8.5), as well as for at least one RCP (8.5) across the selected ESMs; (iii) separate the climate from the fishing signal by including simulations with and without fishing in historical and future runs; (iv) and for regional models, separate the effect of running the model with local data (key run, which may use statistically downscaled inputs) compared to data subset from a global ESM. The core simulations from the initial round are:

*Historical runs* – For historical runs for both global and regional models, the top priority (Tier 1) was one run each with all the climate data sets (reanalysis-based, CMIP5-based) with default settings for fishing effort/mortality (= time-varying effort/mortality) and ocean acidification (= time-varying pH). The lower priority (Tier 2) was to run the same set of climate data with no fishing (= zero fishing effort/mortality) as a sensitivity experiment (see Table 6).

*Future runs* – For future runs with both global and regional models, the top priority (Tier 1) was one run each with all the climate data sets (reanalysis-based, CMIP5-based) with default settings for fishing effort/mortality (= constant at 2005 levels) and ocean acidification (= time-varying pH) for both the RCP2.6 and 8.5 scenarios. The second priority (Tier 2) was to run the same set of climate data with no fishing (= zero fishing effort/mortality) as a sensitivity experiment (see Table 6). The third priority (Tier 3) was the full set of experiments for RCP4.5 and 6.0 scenarios where available (at present, only the IPSL-CM5A-LR model).

For all runs, it was requested that all non-specified external forcings (e.g. habitat modification) should be kept at default settings (time-varying until 2005, constant at 2005 levels into the future). Input data were provided from 1951/1959 to 2004/2005, with a request that years until 1970 should be replicated as needed and used for spin-up (spin-up to be decided individually by each modelling group). Historical reporting was from 1971-2005, or whenever the model started if later.

The Fish-MIP v1.0 protocol will necessarily be revised and revisited, as new climate and fishing data become available, as new ecosystem models are included within Fish-MIP (which is encouraged), as shared understanding of approaches to marine ecosystem modeling increases and as existing models evolve. The most recent protocol for ISIMIP simulation round 2A is available at: https://www.isimip.org/protocol/#isimip2A.

## 6. Conclusions

We believe that the broad intercomparison of marine ecosystem models facilitated by Fish-MIP provides a useful step towards improving our understanding of the future of the marine realm and catalyzing development and uptake of these models. The wide diversity of marine ecosystem models provides a healthy spread of perspectives on what are ultimately



very complex biological and ecological systems, and may provide insight into critical processes that may be incorporated in only a subset of models. We expect that, as in other sectors, model intercomparison will help identify processes that are under- or misrepresented in individual models or model types, and spur model improvement. Here we have described the Fish-MIP project and protocol in preparation for forthcoming model comparisons at multiple scales. We hope that material

compiled for Fish-MIP will inform other intercomparison projects and drive interactions between marine ecosystem modelers and those working in other disciplines. Several marine ecosystem models included in Fish-MIP have already supported projections of the future state of the seas and climate impacts on fisheries. The Fish-MIP intercomparison will add to this by systematically highlighting the uncertainty associated with different model structures and assumptions. This will ultimately improve our capacity to convey limitations of any advice on future states of marine ecosystems and fisheries and

to quantify the benefits and risks associated with alternate management, adaptation or mitigation options.

## 7. Code and data availability

The experimental protocol has no code associated with it. The protocol is described in this manuscript, the supplementary material, and can also be downloaded from https://www.isimip.org/protocol/#isimip2a (for simulation round 2A) and https://www.isimip.org/about/isi-mip2/fisheries/ (will include any updates or additions). The Fish-MIP website is

https://www.isimip.org/gettingstarted/marine-ecosystems-fisheries/. Forcing data from CMIP5 used for the Fish-MIP simulation round 2A are available on the ISI-MIP servers (https://www.isimip.org/gettingstarted/#how-to-join-isimip); fisheries forcing data for specific fished model runs and models are available by contacting individual Fish-MIP modelling groups. Fish-MIP model outputs from simulation round 2A will be made publicly available on the ESGF server (with associated DOI) in January 2018.

*Author contributions.* DPT, TDE, HKL, EDG, WC, JS and VH led the development of the protocol, with contributions from the other authors. DPT led the writing of the text. All authors contributed to the text.

*Competing interests:* The authors declare that they have no conflict of interest.

*Acknowledgements.* Financial support was provided by the German Federal Ministry of Education and Research (BMBF, grant no. 01LS1201A1) through the Inter-Sectoral Impact Model Inter-comparison Project (ISIMIP). We thank Reg Watson for supporting CMIP5 data preparation, and John Dunne and Keith Lindsay helping provide us with access to ESM outputs. DT thanks the Kanne Rassmussen Foundation, Denmark, and the Cambridge Conservation Initiative grant CCI-05-14-018

for financial support that facilitated work on this paper. Additional financial support for the CMIP5 input data preparation was provided by the Australian Research Council Discovery project (DP140101377). HKL acknowledges financial support from the Natural Sciences and Engineering Research Council (NSERC) of Canada. JB acknowledges the UK Natural





Environment Research Council and Department for Environment, Food and rural Affairs [grant number NE/L003279/1]. SN acknowledge support from the NordForsk-funded project Green Growth Based on Marine Resources: Ecological and Socio-Economic Constraints. MC was partially funded by the European Commission through the Marie Curie Career Integration Grant Fellowships [PCIG10-GA-2011-303534] – to the BIOWEB project. SJ acknowledges support from the UK

Department of Environment, Food and Rural Affairs. YJS and ROR were partly funded through the EMIBIOS project (FRB Fondation pour la Recherche sur la Biodiversité, Contract No. APP-SCEN-2010-II). ROR thanks the financial support from the IMARPE-PRODUCE-IADB Project "Adaptation to climate change of the fishery sector and marine-coastal ecosystem of Perú" (PE-G1001/PE-T1297). BF acknowledges funding from both the Fisheries Research and Development Corporation and CSIRO.

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



**Table 1:** A taxonomy of marine ecosystem models taking part in the Fish-MIP project. See also Tables 2 and 3 for the degree of heterogeneity in inputs and outputs that also exists across the model types.

| Fish-MIP model | Brief model description | Domain | Defining features and key processes | Spatial and temporal scale and vertical resolution | Taxonomic scope | Key reference |
|---|---|---|---|---|---|---|
| **SPECIES DISTRIBUTION MODELS** - *statistical relationships between species and environment. Focus on role of habitat change and population dynamics* | | | | | | |
| DBEM | The DBEM defines a bioclimatic envelope for each species, and simulates changes in abundance and carrying capacity under a varying environment. | Global | Carrying capacity is a function of the environment and species' preferences for these factors. Movement of adults is driven by a gradient of habitat suitability and density. Larval dispersal is dependent on currents and temperature. Growth, reproduction and mortality are dependent on oxygen, pH and temperature | ½ × ½ degree; model outputs are annual average. Vertical dimension implicit through species niche preferences. | Fish and invertebrate species (primarily commercial) | (Cheung et al., 2011) |
| SS-DBEM | SS-DBEM is based on the DBEM and the macroecological model, and projects changes in species distribution, abundance and body size, and includes populations dynamics, dispersal, competition. | Global | Key processes include ecophysiology, population dynamics, dispersal, trophic interactions, fishing mortality, and habitat suitability | ½ × ½ degree and yearly. Often aggregated into management or ecological meaningful units (e.g. EEZ, LMEs or ICES areas). Vertical dimension implicit through species niche preferences. | All trophic levels of fish and invertebrates. | (Fernandes et al., 2013) |
| **TROPHODYNAMIC MODELS** – *structured based on species interactions and transfer of energy across trophic levels* | | | | | | |
| Ecopath with Ecosim (EwE) | Ecopath with Ecosim is a mass-balance food web model that accounts for the flow of biomass between trophic groups. | Regional | Includes a mass-balance component (Ecopath), a temporal dynamic component (Ecosim) and a spatial-temporal dynamic component (Ecospace). Typically resolved to a mix of functional groups and key species. | Spatial resolution varies from local to global, gridded configuration. Flexible, typically running in monthly time steps. Depth dimension is considered implicitly through food web interactions and habitat preference pattern. | All trophic levels and taxonomic groups can be included as biomass pools or age-structured life history stanzas. | (Christensen and Walters, 2004) |

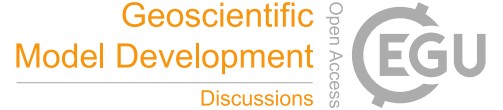



| Fish-MIP model | Brief model description | Domain | Defining features and key processes | Spatial and temporal scale and vertical resolution | Taxonomic scope | Key reference |
|---|---|---|---|---|---|---|
| EcoOcean | EcoOcean is a global food web model based on the EwE framework, designed to evaluate the impact of climate change and human pressure on marine ecosystems. | Global | Atmosphere–ocean circulation model (COBALT); EwE food web model with Ecosim and Ecospace habitat capacity model; fisheries effort global gravity model | Global model. Spatial resolution is ½ × ½ degree and outputs are annual or monthly averages Depth dimension is considered implicitly through food web interactions and habitat preference pattern. | All trophic levels and taxonomic groups included as biomass pools (51 groups) | (Christensen et al., 2015) |
| *SIZE-BASED MODELS* – developed from food-web, macroecological, and life-history theory for exploration of community size-spectra | | | | | | |
| Macroecological model | A static model, which uses minimal inputs together with ecological and metabolic scaling theory to predict mean size composition and abundance of animals (including fish). | Global | Provides a simple size-based characterisation of marine ecosystems. Relies on estimates of predator-prey mass ratios, transfer efficiency and changing metabolic demands with body mass and temperature to predict body mass distributions and abundance of marine consumers from phytoplankton primary production and environmental temperature. Ignores non-phytoplankton production and animal movement. | Static equilibrium model, typically applied at scales from 0.5×0.5 degree grids to large marine ecosystems; forced with annual or monthly mean environmental variables. Single vertical (surface-integrated) layer. | Species are not resolved, only body mass classes | (Jennings and Collingridge, 2015) |
| Dynamic Pelagic-Benthic Model (DPBM) | A functional trait-based size spectrum model that joins a pelagic predator size-spectra model with a benthic detritivore size spectrum; can include herbivores or other groups that do not feed according to size and unstructured resources. | Global or regional | Individual processes of predation, food dependent growth, mortality and reproduction give rise to emergent size spectra for each functional group. Can be linked to GCMs, regional models or observations via parameterizing phyto- and zooplankton size-spectra, detritus and or temperature. | Spatial scale of grid is flexible and dependent on inputs; temporal scale daily or weekly; two vertical layers (sea-surface and sea-floor. | Broadly represents "pelagic" fish predators, "benthic" invertebrates but can include herbivorous fish; flexible functional groups. | (Blanchard et al., 2012b) |
| BOATS | Combines size-based ecological theory and metabolic constraints to calculate the production of fish, resolved across multiple size spectra, and applies a coupled | Global or regional | Applies empirical parameterizations to describe phytoplankton community structure, trophic transfer of primary production from phytoplankton to fish, growth rates, and natural mortality. Model parameters are calibrated against observed using a Monte Carlo | Flexible spatial scale; typically global, at 1×1 degree; monthly timestep; single vertical (surface-integrated) layer. | All commercial species represented by three groups, defined in terms of the asymptotic | (Carozza et al., 2016) |





| Fish-MIP model | Brief model description | Domain | Defining features and key processes | Spatial and temporal scale and vertical resolution | Taxonomic scope | Key reference |
|---|---|---|---|---|---|---|
| | economic model to determine effort and harvest based on economic boundary conditions. | | technique. Explicitly models the evolution of effort and harvest. Recruitment is dependent on stock size and the environment, and simple life history features are resolved. | | mass. | |
| POEM | A mechanistic ecosystem model that uses body-size as the basis of interaction. Offline coupled to an Earth System Model, using zooplankton biomass and mortality fields to force ecosystem dynamics. | Global | Simple size-based relationships defined by empirical allometric relationships are used to model ecological interactions. | Spatial scale 1×1 degree; daily timestep; single vertical layer representing upper 200m | One or two "Ecotypes", e.g. a piscivore or a planktivore | (Watson et al., 2015) |
| *COMPOSITE (HYBRID) MODELS* – *including multiple (e.g. size, age, trophic, physical, and other) model formulations in system representation* | | | | | | |
| Atlantis | Atlantis is a whole ecosystem model, taking a transport model derived from hydrodynamic or GCM output that sets the conditions for a full representation of the food web and human users. | Regional | Modular (multiple options per process). Includes age structure and major ecological processes such as full life history closure, gape-limited predation, habitats, movement, biogeochemical nutrient cycling and a range of effort allocation options. | 3D spatial polygons matched to biophysical features; vertically resolved using "slab" layers (with finer layers and the surface and thicker at depth). Timestep is flexible, typically 6-24 hrs. | All trophic levels and taxonomic groups can be represented using a mix of biomass pools and age structured populations. Typically resolved to a mix of functional groups and key species. | (Fulton et al., 2011a) |
| OSMOSE | The higher trophic level model OSMOSE (Object-oriented Simulator of Marine ecOSystems Exploitation) is a spatial multispecies and individual-based model which focuses on fish species. Its current structure embeds a coupling with hydrodynamic and biogeochemical models. | Regional | Trophic interactions are size-based so the modelled food-webs are dynamic. The whole life cycle of the modelled species is represented (migration, food-dependent growth, reproduction and mortality), with tracking of all life stages (from eggs to terminal age) in space and time. Provides size-, age-, species-, trophic level-based indicators in output. | Flexible. Typically, resolution of 1/6 degree and a weekly time-step. Spatially resolved in 2D; the vertical distribution of species is handled through a matrix of accessibility. | Fish and invertebrate species and functional groups | (Travers et al., 2009) |





| Fish-MIP model | Brief model description | Domain | Defining features and key processes | Spatial and temporal scale and vertical resolution | Taxonomic scope | Key reference |
|---|---|---|---|---|---|---|
| SEAPODYM | SEAPODYM is an Eulerian modelling framework including functional groups of lower and mid-trophic levels and populations dynamics of target species, developed for investigating spatial pelagic fish populations dynamics under the influence of fishing and environment | Regional or global | Functional groups of zooplankton and micronekton are simulated and used with physical and biogeochemical variables to define the habitats, movements and key population dynamics processes of targeted fish species. Fishing impact is simulated through catch and effort data. A statistical optimization approach uses all available data (catch by size, tagging data, larvae density, acoustic estimates) to estimate model parameters. | Flexible; typically 1/12 degree grid and daily timestep, or 1-2 degree grid x monthly time-step. 3 vertical layers of epi- and mesopelagic ocean. | One zooplankton and several micronekton functional groups defined based on their vertical behaviour and one to several targeted species with their fisheries. | (Lehodey et al., 2008; 2010) |
| APECOSM | A 3D dynamic energy budget-based Eulerian model of size structured marine populations and communities, based on environmentally-driven individual bio-energetics, trophic interactions and behaviours that are up-scaled to populations and communities. | Regional or global | Includes light and temperature-driven size-based predation, food and temperature-driven growth, reproduction and senescence, impact of the environment on vertical and horizontal movements as well as schooling. | Can be run on any 3D spatial grid from regional to global scale, with a daily timestep distinguishing day and night; no vertical resolution but vertical movements explicitly parameterized. | Generic size-based communities are explicit (typically epipelagic, migratory, mesopelagic and bathypelagic) as well as focus species. | (Maury, 2010) |
| Madingley | A global, mechanistic, spatially-explicit 'general ecosystem model' of terrestrial and marine ecosystems, used to explore changes in ecosystem structure and function. | Regional or global | Models functional groups and multi-species 'cohorts'; millions of cohorts in model. Includes spatially explicit dispersal driven by ESM outputs, food-dependent growth, starvation and senescence mortality. Allows for complete extinction of functional groups, dynamic changes in ecosystem structure. Food web links are dynamic; cohorts can be prey or predator depending on size. Unlike many models, is not 'mass-balanced'. Predators can switch between prey groups based on densities and preferences. | Any (typically 1-2 degrees & monthly), with no vertical resolution at present | All (marine and terrestrial) excluding microbes, modelled as functional groups. | (Harfoot et al., 2014) |



**Table 2.** Selected Earth system model outputs that are required or optional for individual Fish-MIP models. Fish-MIP models can require surface values; a mean or summed surface layer value (e.g. top 100m); surface and seafloor values; or fully three-dimensional values. Note that this list is correct at time of writing, but that models are continuously in development and new components and requirements being added. Units follow CMIP5 standard output (http://cmip-pcmdi.llnl.gov/cmip5/data_description.html).

| | COMMON VARIABLES USED BY AT LEAST 50% OF Fish-MIP MODELS (provided for all Fish-MIP simulations) | | | | | | | SELECTED VARIABLES USED BY A SMALL PROPORTION OF MODELS | | | | |
|---|---|---|---|---|---|---|---|---|---|---|---|---|
| | u & v current speed $[m\ s^{-1}]$ | Sea temperature or potential temperature $[K]$ | Dissolved $O_2$ concentration $[mol\ m^{-3}]$ | Primary organic carbon productivity $[mol\ m^{-3}\ s^{-1}]$ | Zooplankton carbon concentration $[mol\ m^{-3}]$ | pH $[unitless]$ | Salinity $[psu]$ | Total alkalinity $[mol\ m^{-3}]$ | Phytoplankton carbon concentration $[mol\ m^{-3}]$ | Turbulent mixing $[m^2\ s^{-1}]$ | Ice coverage $[fraction]$ | Mixed-layer depth $[m]$ |
| DBEM | • | • | • | | • | • | • | | | | • | |
| SS-DBEM | • | • | • | • | | • | • | | | | • | (•) |
| Ecopath With Ecosim | (•) | (•) | (•) | • | • | (•) | (•) | | • | | | |
| EcoOcean | (•) | • | (•) | • | (•) | (•) | (•) | | • | | (•) | |
| Macroecological model | | • | | • | (•) | | | | (•) | | | • |
| DPBM | | • | | (•) | (•) | | | | (•) | | | • |
| BOATS | | • | | • | | | | | | | | |
| POEM | • | • | | | • | • | • | | | | • | • |
| Atlantis | • | • | • | Ω | † | • | • | • | Ω | • | • | |
| OSMOSE | | • | • | • | •# | | • | | •# | | | |
| SEAPODYM | • | • | • | • | | | | | • | | | |
| APECOSM | • | • | • | | •# | • | | | •# | (•) | | |
| Madingley | • | • | | • | † | | • | | | | | |

• Used by model

(•) Can optionally be used by model

† Not used for forcing, but can be used for cross-validation

Ω Not used directly, but in combination with hydrodynamic flows is used to set boundary conditions and sub-grid scale processes to allow for



similar primary productivity shifts as in ESM outputs to be realized.

\# Separated into large and small size-classes





**Table 3.** Earth system model derived forcing variables provided as input for global and regional marine fisheries models. Names and units follow CMIP5 standard output (http://cmip-pcmdi.llnl.gov/cmip5/data_description.html).

| Variable | Name | Unit (assuming depth-resolved) | Frequency | Comments |
|---|---|---|---|---|
| u current | *uo* | $m\ s^{-1}$ | Monthly | |
| v current | *vo* | $m\ s^{-1}$ | Monthly | |
| Temperature | *t* | $K$ | Monthly | |
| Dissolved oxygen concentration | o2 | $mol\ m^{-3}$ | Monthly | |
| Primary organic carbon productivity | *intpp* | $mol\ m^{-3}\ s^{-1}$ | Monthly | Sum of primary productivity by all primary producers (3 groups – lphy, sphy, diaz for GFDL-reanalysis and GFDL-ESM2M, 2 groups – lphy, sphy for IPSL) |
| Phytoplankton carbon concentration | *phyc* | $mol\ m^{-3}$ | Monthly | Sum of small and large phytoplankton (including diazotrophs) |
| Small phytoplankton carbon concentration | *sphyc* | $mol\ m^{-3}$ | Monthly | Pico- and nano-phytoplankton |
| Large phytoplankton carbon concentration | *lphyc* | $mol\ m^{-3}$ | Monthly | Diatoms, large non-diatoms, and diazotrophs |
| Zooplankton carbon concentration | *zoo* | $mol\ m^{-3}$ | Monthly | Sum of small and large zooplankton |
| Small (micro)zooplankton carbon concentration | *szoo* | $mol\ m^{-3}$ | Monthly | Post-diagnosed by normalizing to phytoplankton where unavailable |
| Large (meso)zooplankton carbon concentration | *lzoo* | $mol\ m^{-3}$ | Monthly | Post-diagnosed by normalizing to phytoplankton where unavailable |
| pH | *Ph* | *unitless* | Monthly | |
| Salinity | *So* | *psu* | Monthly | |





**Table 4.** Selected outputs produced by individual models. Note that this list is correct at time of printing, but that models are continuously in development and new components and requirements being added. This table lists a range of potential outputs from models participating in Fish-MIP; for the list of requested and optional model outputs see Table 5.

| | COMMON OUTPUTS PRODUCED BY AT LEAST 50% OF MODELS | | | | | | | OPTIONAL OUTPUTS PRODUCED BY A SMALL PROPORTION OF MODELS | | |
|---|---|---|---|---|---|---|---|---|---|---|
| | Fish species / functional group carbon biomass density [g m⁻³ month⁻¹] | Fisheries metrics [various] | Relative species / functional group abundances [unitless] | Trophic level [unitless] | Production of carbon [g m⁻³ month⁻¹] | Production / biomass ratio [unitless] | Mortality rate [month⁻¹] | Species / functional group diversity [unitless] | Individual-based metrics (e.g. growth rates) [various] | Food-web interaction metrics [various] |
| DBEM | | • [4] | • | | | | | | | |
| SS-DEBM | • [1] | • [5] | • | | | | | | | |
| Ecopath with Ecosim | • | • [6] | • | • | • | • | • | • | | • |
| EcoOcean | • | • [7] | • | • | • | • | • | • | | • |
| Macroecological model | • | | | • | • | • | • | | | |
| DPBM | • [1] | • [6,8] | • | • | • | • | • | | • | • |
| BOATS | • [1] | • [6] | | • | • | • | • | | • | |
| POEM | • | | | | • | | | | | |
| Atlantis | • | • [6] | • | • | • | • | • | • | • | • |
| OSMOSE | • | • [6] | • | • | • | • | • | • | • | • |
| SEAPODYM | • [2] | • [9] | | | • | • | • | | | |
| APECOSM | • [3] | • [10] | | • | | | • | • [10] | • | |
| Madingley | • | | • | • | | | | • | • | • |

1. As a size-spectrum

2. Tunas and associated species, age-structured





3. Size-spectrum for epipelagic, migratory, meso-pelagic and bathypelagic communities, together with focus species (e.g., tunas)

4. Relative functional group abundances

5. Catch-rates

6. Catch and fishing mortality

5  7. Seafood production

8. Potential catch

9. Catch & size-frequency for tunas

10. Commercial landings





**Table 5.** Common output variables to be provided by global and regional marine fisheries models.

| Output variable | Variable name | Resolution | Unit (NetCDF format) | Comments |
|---|---|---|---|---|
| Total system carbon biomass | *tsb* | Monthly | $g\ m^{-2}$ | All primary producers and consumers |
| Total consumer carbon biomass density | *tcb* | Monthly | $g\ m^{-2}$ | All consumers (trophic level >1, vertebrates and invertebrates) |
| Carbon biomass density of consumers >10cm | *b10* | Monthly | $g\ m^{-2}$ | If asymptotic length ($L_{inf}$) is >10cm, include in >10cm class |
| Carbon biomass density of consumers >30cm | *b30* | Monthly | $g\ m^{-2}$ | If asymptotic length ($L_{inf}$) is >30cm, include in >30cm class |
| Total catch (all commercial functional groups / size classes) | *tc* | Monthly | *g wet biomass $m^{-2}$* | Catch at sea (commercial landings plus discards), fish and invertebrates. Fished runs only. |
| Total landings all commercial functional groups / size classes) | *tla* | Monthly | *g wet biomass $m^{-2}$* | Commercial landings (catch without discards), fish and invertebrates. Fished runs only. |

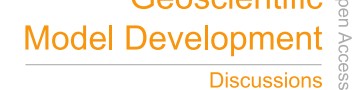



**Table 6.** All experiments (historical and future, standardized and optional) for the global and regional fisheries and marine ecosystem models participating in the first round of Fish-MIP. Runs in dark text are prioritized (Tier 1), those in grey preferred but optional (Tier 2) and those in grey and italic optional (Tier 3); this is to allow modellers with limited computational resources to participate and prioritize. Note that the CMIP5-based runs are continuous from historical into the future, reducing the total number of runs.

| Earth system model forcing | Scenario | Time period | Fishing effort | Ocean acidification | # runs |
|---|---|---|---|---|---|
| GFDL ESM2M (re-analysis) | historical | 1971-2005 | default (time-varying effort/mortality) <br> unfished (zero effort/mortality) | default (time-varying pH) | 2 |
| IPSL-CM5A-LR | historical | 1971-2005 | default (time-varying effort/mortality) <br> unfished (zero effort/mortality) | default (time-varying pH) | 2 |
| GFDL ESM2M | historical | 1971-2005 | default (time-varying effort/mortality) <br> unfished (zero effort/mortality) | default (time-varying pH) | 2 |
| IPSL-CM5A-LR | 2.6 (rcp2p6) <br> 8.5 (rcp8p5) | 2006-2099 | keep constant at 2005 levels <br> unfished (zero effort/mortality) | default (time-varying pH) | 4 |
| *IPSL-CM5A-LR* | *4.5 (rcp4p5)* <br> *6.0 (rcp6p0)* | *2006-2099* | *keep constant at 2005 levels* <br> *unfished (zero effort/mortality)* | *default (time-varying pH)* | *4* |
| GFDL ESM2M | 2.6 (rcp2p6) <br> 8.5 (rcp8p5) | 2006-2099 | keep constant at 2005 levels <br> unfished (zero effort/mortality) | default (time-varying pH) | 4 |