# Peer review of "A protocol for the intercomparison of marine fishery and ecosystem models: Fish-MIP v1.0"

_Geoscientific Model Development, 2017_

## Referee Comment (RC1) · Anonymous Referee #1 · 8 Nov 2017

General comments:

This work develops a model inter-comparison protocol for fisheries and ecosystem models forced by similar climate scenarios. Achieving this is important in order to better understand and evaluate predictions coming from these models, and difficult due to the diversity of model structures, scales, inputs, and outputs. This intercomparison protocol is novel; comparing these diverse model types has not been attempted before. However, it builds upon the existing coupled and ocean model inter-comparison projects (CMIP and OMIP), and is part of the Inter-Sectoral Impact Model Intercomparison Project, a broader effort. The manuscript is well-written, well-referenced, and clear, with a descriptive title and abstract.

This work represents a substantial advance in modelling science. In concept, both the

development of the protocol and the subsequent intercomparisons will facilitate standardization and improvements to multiple types of ecosystem and fisheries models, as well as allow ensembles of these models to be assembled in a systematic way. This will happen even as the protocol is improved, as noted below in specific comments. While this is the first version of the protocol and comparative results are not presented, one interesting result of employing the protocol is the limited number of earth system models (ESMs) available to force the models after screening for availability, quality control, future response, and model drift. This alone indicates a need for improvement of ESM outputs and potentially fosters two-way communication between modeling research groups.

The two outstanding issues with the manuscript are the validity of the assumption of comparable fishery production using the current protocol, and reproducibility of the protocol as written for individual models. These are already recognized but could be addressed further with minor revisions.

Specific comments:

The methods and assumptions are clearly outlined and mostly valid. The impact of fishing may be difficult to see in projections with the current protocol, however, due to the decision to either hold constant 2005 fishing levels or have no fishing. Comparing 2005 fishing in a lightly fished system with 2005 fishing in a heavily fished system will be difficult; fishery productivity under these circumstances is not comparable. This is clearly a placeholder to get the project up and running, and the authors acknowledge that it is not ideal. I understand how difficult it is to predict future fishing and also how difficult it is to estimate things like dynamic sustainable yield or other target reference points, but I think it is essential to develop protocols for this to have better model intercomparisons in the future. Perhaps the authors could outline in more detail how this might be addressed in future protocols (e.g. how might the future fishing scenarios referenced on p 14, line 14 be made spatially explicit in a standardized way?).

Precisely reproducing the protocol for an individual model may be difficult, because in this first round there seems to be much leeway given to individual modeling groups to "optimize" inputs from forcing models, rather than use standardized inputs. While it makes sense to allow flexibility given the wide range of ecosystem and fisheries models included and the diversity of their inputs, different modeling groups working with the same model framework may make different decisions under the current flexible protocol. The authors note that some standardization will be imposed in updated versions of the protocol. Perhaps additional examples of standardizations of the protocol could be given (one regarding diazotrophs is given on p 11, lines 21-24), or some method for standardizing decision making of modeling groups regarding forcing inputs could be outlined to ensure that similar "optimizations" would be made across groups under the current protocol.

Technical corrections: Define CMIP5 and CMIP6 in abstract P 7, line 18, should this say "Ecospace, the spatial-temporal model run in Ecosim. . ."

---

## Referee Comment (RC2) · K. Rose (Referee) · 15 Nov 2017

This manuscript presents a framework for comparing marine fisheries and ecosystem models. This is clearly an important topic, given the great diversity of modeling approaches, general lack of standard process formulations and documentation, and high role of modeler decisions in model development and implementation. Previous large-scale comparisons mainly focused on paper-based comparisons, while this manuscript lays out an approach for quantitative comparisons. Thus, the topic of the manuscript is of wide interest and broad utility, and the manuscript is well organized and well-written.

I will focus on my major comment because it is substantial in nature and, in my opinion, needs to be addressed for the manuscript to be published and for the manuscript to

maximize its impact. It appears (page 16) that the protocol was actually implemented for multiple models in historical and future runs, but no results are presented. Indeed, the supplemental material has one figure (Figure S4) showing one output for 3 of the models. Furthermore, there is no discussion in the manuscript on lessons learned or potential issues or guidelines when the proposed protocol is actually implemented. In my opinion, the lack of presentation of at least a demonstration that the protocol can be implemented is a major missing aspect of the manuscript. Comparing multiple models, especially with the great diversity of models as accommodated in the proposed protocol, is very much dependent on the details of the implementation. Protocols that propose averaging input values from a common source for the different models and comparing common outputs (a much oversimplified description of the approach used in the manuscript) are intuitive and appear viable. It is when the protocol is actually attempted to be used with actual models and specific scenarios that implementation issues and other details emerge. Thus, the proposed protocol sounds good in theory, but I would suggest that the existing text can be shortened and a new section that demonstrates that the protocol can be effectively implemented be added to the manuscript.

It seems such results for a demonstration example are already available based on the text in the manuscript. One does not need to add an example with all of the models under many scenarios. A demonstration that uses 3-5 models (strategically selected) that cover the major model types (species-distribution, trophodynamic, size or age-based, composite) and spatial scales (global, regional) for 2 scenarios would be sufficient to show the reader that the protocol can actually be implemented and useful comparative results obtained. The actual results of the models are less important than showing the models can be usefully compared using the protocol.

Inclusion of a demonstration will move the manuscript from a proposed protocol (albeit well thought out and presented) to a protocol whose results and approach would much more likely be used by others. I encourage the authors to do this because a protocol,

like the one proposed, is desperately needed to ensure the information generated from models is robust and effectively conveyed among research groups and to managers.

---

## Author Comment (AC1) · 23 Dec 2017

**Response to reviewer comment 1:**

General comments: This work develops a model inter-comparison protocol for fisheries and ecosystem models forced by similar climate scenarios. Achieving this is important in order to better understand and evaluate predictions coming from these models, and difficult due to the diversity of model structures, scales, inputs, and outputs. This intercomparison protocol is novel; comparing these diverse model types has not been attempted before. However, it builds upon the existing coupled and ocean model inter-comparison projects (CMIP and OMIP), and is part of the Inter-Sectoral Impact Model Intercomparison Project, a broader effort. The manuscript is well-written,

well-referenced, and clear, with a descriptive title and abstract.

*Thank you.*

This work represents a substantial advance in modelling science. In concept, both the development of the protocol and the subsequent intercomparisons will facilitate standardization and improvements to multiple types of ecosystem and fisheries models, as well as allow ensembles of these models to be assembled in a systematic way. This will happen even as the protocol is improved, as noted below in specific comments. While this is the first version of the protocol and comparative results are not presented, one interesting result of employing the protocol is the limited number of earth system models (ESMs) available to force the models after screening for availability, quality control, future response, and model drift. This alone indicates a need for improvement of ESM outputs and potentially fosters two-way communication between modeling research groups.

*We agree with the reviewer that the limited number of ESMs available to force the models (or, specifically, the limited availability of archived ESM outputs with the necessary three-dimensional forcing variables) indicates a need for improved communication between earth-system modellers and marine ecosystem modellers, and we are pleased to report that this process is underway and gaining momentum. In relation to the comment that "comparative results are not presented", please note that we have now added example results in response to a comment by the second reviewer.*

The two outstanding issues with the manuscript are the validity of the assumption of comparable fishery production using the current protocol, and reproducibility of the protocol as written for individual models. These are already recognized but could be addressed further with minor revisions.

*These issues are addressed in our responses to specific comments below.*

[Figure]

Specific comments: The methods and assumptions are clearly outlined and mostly valid. The impact of fishing may be difficult to see in projections with the current protocol, however, due to the decision to either hold constant 2005 fishing levels or have no fishing. Comparing 2005 fishing in a lightly fished system with 2005 fishing in a heavily fished system will be difficult; fishery productivity under these circumstances is not comparable. This is clearly a placeholder to get the project up and running, and the authors acknowledge that it is not ideal. I understand how difficult it is to predict future fishing and also how difficult it is to estimate things like dynamic sustainable yield or other target reference points, but I think it is essential to develop protocols for this to have better model intercomparisons in the future. Perhaps the authors could outline in more detail how this might be addressed in future protocols (e.g. how might the future fishing scenarios referenced on p 14, line 14 be made spatially explicit in a standardized way?).

*We agree that holding fishing levels constant at 2005 levels in the future projections was unrealistic, but the development of such future projections remains a significant community-wide research project, and making this assumption allowed us to progress the simulations in the current absence of more refined projections. We certainly agree that improved scenarios are necessary to better quantify the projected impacts of fishing, and have followed the reviewer's suggestion that we could outline in more detail how this might be accomplished. To this end, we have added the following to the manuscript on page 14:*

*"Developing more sophisticated future scenarios of fisheries remains challenging. Formally-developed global qualitative storylines depicting future fishing activity, management, and technological change are beginning to be designed (e.g. Maury et al., 2017), but need to be 'operationalized' in terms of translating them into a spatially and temporally explicit form to enable them to force marine ecosystem models. The development of such global projections of fishing pressure over the 21st century will*

*be necessary to better understand the consequences of interactions between climate and fisheries effects on ecosystems and resultant yields. Deriving projections that recognize the complexities of fisheries management remains particularly challenging, even at regional scales which may more naturally map to specific existing management units. Such projections would need to account for the significance of many drivers and feedbacks which influence mortality rates and the catches that result. These include economic drivers of fleet capacity, effort and distribution; environmental and fisheries policies and associated management targets, the extent to which targets are met and their responsiveness to changes in the environment, fisheries and society; the selectivity and efficiency of fishing operations; fishery and species interactions; and external modifiers of demand for, and access to, wild fish for food (e.g. growth of aquaculture, marine conservation, certification, ethics). A further challenge is to develop comparable projections for different fish production models: for example, with appropriate assumptions necessary to translate between species and size-based projections of mortality and selection. The underlying difficulty with developing all such projections is that yield is not a stable construct, but changes dynamically in relation to the species and sizes targeted, with feedbacks, and with the evolution of the fished ecosystem. When future fishing scenarios do become operationalized, they will be used to replace the standardized run with constant 2005 catch or effort in future model experiment protocols."*

Precisely reproducing the protocol for an individual model may be difficult, because in this first round there seems to be much leeway given to individual modeling groups to "optimize" inputs from forcing models, rather than use standardized inputs. While it makes sense to allow flexibility given the wide range of ecosystem and fisheries models included and the diversity of their inputs, different modelling groups working with the same model framework may make different decisions under the current flexible protocol. The authors note that some standardization will be imposed in updated versions of the protocol. Perhaps additional examples of standardizations of the protocol could

be given (one regarding diazotrophs is given on p 11, lines 21-24), or some method for standardizing decision making of modelling groups regarding forcing inputs could be outlined to ensure that similar "optimizations" would be made across groups under the current protocol.

*This is an excellent point. There are two separate effects here: firstly, the effect of modellers using the same modelling framework (e.g. Ecopath with Ecosim) but making different configuration choices. Note that this primarily affects regional models, since global modellers all use different modelling frameworks. Secondly, there is the effect of the incorporation or lack of incorporation of specific ecosystem facets or processes, which can affect all models. With regard to the former, we note that regional models are often specifically fitted to regional data to best capture observed historical trends in data for that particular region, and we prefer to maintain this approach. Furthermore, as we do not (yet) have two versions of a specific ecosystem model type (e.g. EwE) in a single region, we do not believe that this is yet an issue. Nonetheless, we agree that capturing specific configuration setups for each model in future protocol iterations should be done in order to better capture the decisions that were made and their potential consequences. We have added the following sentence to the manuscript (p16):*

*"While we did not have any overlap in terms of different modelling groups using the same model or software framework in a specific region (or at global scales), to tackle this situation in future Fish-MIP protocols, and specifically the potential for different modelling groups to make differing configuration decisions, we plan to update our documentation associated with each simulation run to include the full configuration choices that were made."*

*With regard to the second effect, the reviewer specifically highlights the inclusion or lack of inclusion of diazotrophs that we documented in the manuscript. This example was in fact a surprise to us, as these differences are not easily foreseeable and only become revealed as the model intercomparison process proceeds. We therefore cannot give another example of standardizing decision making (beyond those in the MS),*

*because we have already included such situations which arose during this process in the v1.0 protocol. However, as noted in the MS, this will be an evolving protocol and process. The experience with model intercomparison projects in other sectors – specifically in ISIMIP (Frieler et al., 2017, GMD) – has shown that with each iteration of protocols and simulations, mutual understanding between modellers increases and more relevant differences in processes and parameters are identified that can then be documented and, if necessary, be harmonized. Equally, the main learning curve here has occurred simply by attempting this project, and we hope and anticipate that identifying such potential differences and issues is one of the outcomes of this project, as is the community buy-in that will make the convening of modellers to identify such problems more feasible.*

Technical corrections: Define CMIP5 and CMIP6 in abstract P 7, line 18, should this say "Ecospace, the spatial-temporal model in Ecosim..."

*We have updated the abstract as follows:*

"The current Fish-MIP protocol is designed to allow these heterogeneous models to be forced with common Earth System Model (ESM) Couple Model Intercomparison Project Phase 5 (CMIP5) outputs under prescribed scenarios for historic (from 1950s) and future (to 2100) time periods; it will be adapted to CMIP phase 6 (CMIP6) in future iterations."

We have also fixed the indicated sentence about Ecopath.

**Response to reviewer comment 2:**

This manuscript presents a framework for comparing marine fisheries and ecosystem models. This is clearly an important topic, given the great diversity of modeling approaches, general lack of standard process formulations and documentation, and high role of modeler decisions in model development and implementation. Previous large

scale comparisons mainly focused on paper-based comparisons, while this manuscript lays out an approach for quantitative comparisons. Thus, the topic of the manuscript is of wide interest and broad utility, and the manuscript is well organized and well-written.

*Thank you.*

I will focus on my major comment because it is substantial in nature and, in my opinion, needs to be addressed for the manuscript to be published and for the manuscript to maximize its impact. It appears (page 16) that the protocol was actually implemented for multiple models in historical and future runs, but no results are presented. Indeed, the supplemental material has one figure (Figure S4) showing one output for 3 of the models. Furthermore, there is no discussion in the manuscript on lessons learned or potential issues or guidelines when the proposed protocol is actually implemented. In my opinion, the lack of presentation of at least a demonstration that the protocol can be implemented is a major missing aspect of the manuscript. Comparing multiple models, especially with the great diversity of models as accommodated in the proposed protocol, is very much dependent on the details of the implementation. Protocols that propose averaging input values from a common source for the different models and comparing common outputs (a much oversimplified description of the approach used in the manuscript) are intuitive and appear viable. It is when the protocol is actually attempted to be used with actual models and specific scenarios that implementation issues and other details emerge. Thus, the proposed protocol sounds good in theory, but I would suggest that the existing text can be shortened and a new section that demonstrates that the protocol can be effectively implemented be added to the manuscript.

It seems such results for a demonstration example are already available based on the text in the manuscript. One does not need to add an example with all of the models under many scenarios. A demonstration that uses 3-5 models (strategically selected) that cover the major model types (species-distribution, trophodynamic, size or age-based,

composite) and spatial scales (global, regional) for 2 scenarios would be sufficient to show the reader that the protocol can actually be implemented and useful comparative results obtained. The actual results of the models are less important than showing the models can be usefully compared using the protocol.

*The reviewer is correct that there are model outputs that could potentially be included into this paper. We had originally elected not to show these since their extraction, plotting, and analysis is a considerable task in-and-of-itself, and this paper is intended to be a stand-alone description of the model protocol, which in itself is fairly substantial. We do have an additional paper in review that begins to tackle the interpretation of the results. Nonetheless, noting the importance that the reviewer placed on showing a demonstration that the protocol is actually feasible and implementable in practice, we have added a figure to the main manuscript that shows some example results. As per the reviewer's suggestion, we show outputs from different model types (species-distribution, trophodynamic, size-based, and composite) for a variety of scenarios (global and regional scales, IPSL and GFDL forcing, fishing and no-fishing scenarios, RCP 2.6 and RCP 8.5).*

*The reviewer suggests that there is no discussion in the manuscript about lessons learned or potential issues, but we suggest that in fact this is included in the MS: issues such as the diazotrophs only emerged when attempting to implement the protocol, as did the necessity for a hierarchy of simulation runs, the post-diagnosing of zooplankton size-classes, and other issues. These are all included in the MS, but presented as part of the current protocol (i.e., here you are seeing the outcome of a near-3-year process of iteratively developing a protocol as issues are found, modellers join, simulations are re-run, and so-forth). Note that in the original version of this manuscript (prior to a revision after which it was then sent for review), we did in fact have something akin to a 'lessons learned' section, and a much broader scope in terms of analysing the process and outcomes of the protocol. However, this did not correspond well to the 'Model experiment description paper' section of GMD, and we were asked to revise*

*the manuscript to bring it in line with this format. We now accept that this 'protocol' description is very appropriate for providing the stand-alone foundation for the Fish-MIP project that can then be cited by all papers that explore model outputs, examine ecological or technical issues, or indeed reflect on lessons learned and improvements for upcoming protocol iterations. Given this focus, we also refrain from shortening the MS or adding a new section on model results, but as the reviewer suggests, instead show that the models can be 'usefully compared', and give some examples of what the output looks like.*

*We have added the following text to the manuscript:*

*Section 5 (Core simulations) page 17:*

*"Figure 1 shows example outputs from regional and global model runs for a subset of the Fish-MIP models. Note that results can be visualised either spatially or as time-series, as absolute values or relative changes, and can be represented as individual simulation outputs for specific models, and/or averages across multiple models. While we refrain from discussing results and specific values, as only a subset of models and simulations are shown here, we do note that there is substantial temporal variation in magnitude and direction of trends between models, and spatial variation in the ensemble model mean. Separate papers from the Fish-MIP project will provide analysis of the full suite of results. Furthermore, all simulation results are being made publicly available (see 'code and data availability' section) to enable the whole community to analyse and interpret results."*

*Figure caption:*

*"Figure 1: Example outputs from the Fish-MIP v1.0 protocol core simulations. A) Global model time-series output. Percentage change in global spatially-averaged total consumer carbon biomass density (g m-2) from 1990 to 2050. All values are relative to the 1990-1999 mean. Values are shown for three marine ecosystem models: a size-based model (BOATS), a trophodynamic model (EcoOcean), and a species*
*distribution model (DBEM). Output only shown for IPSL RCP 8.5 model runs without fishing imposed. For definition of total consumer carbon biomass see Table 5. B) Regional model time-series output. Percentage change in spatially-averaged total consumer carbon biomass density (g m-2) from 1990 to 2050. All values are relative to the 1990-1999 mean. Values are shown for two regional marine ecosystem models in southeast Australia: a trophodynamic model (Ecopath with Ecosim) and a composite (hybrid) model (Atlantis). Spatial extent of models is overlapping but non-identical. Output only shown for IPSL RCP 2.6 and IPSL RCP 8.5 model runs without fishing imposed. C) Global spatial model output runs (models as per panel A) for IPSL RCP 8.5 with fishing imposed. Ensemble model mean percentage change in total consumer carbon biomass density (g m-2) from the 1990s to the 2050s; a positive value indicates an increase over time. Percentage changes in each grid cell for the three models in panel (A) over this time-period were averaged. See Section 3.2 for details of fisheries forcing."*

Inclusion of a demonstration will move the manuscript from a proposed protocol (albeit well thought out and presented) to a protocol whose results and approach would much more likely be used by others. I encourage the authors to do this because a protocol like the one proposed, is desperately needed to ensure the information generated from models is robust and effectively conveyed among research groups and to managers.

*We hope that the reviewer will recognize that we have addressed their concern through including demonstration results, and hope that, as they suggest, this is likely to increase the uptake of the protocol and intercomparison results.*
* * *
[Figure]

**Figure 1: Example outputs from the Fish-MIP v1.0 protocol core simulations.** A). Global model time-series output. Percentage change in global spatially-averaged total consumer carbon biomass density (g m$^{-2}$) from 1990 to 2050. All values are relative to the 1990-1999 mean. Values are shown for three marine ecosystem models: a size-based model (BOATS), a trophodynamic model (EcoOcean), and a species distribution model (DBEM). Output only shown for IPSL RCP 8.5 model runs without fishing imposed. For definition of total consumer carbon biomass see Table 5. B) Regional model time-series output. Percentage change in spatially-averaged total consumer carbon biomass density (g m$^{-2}$) from 1990 to 2050. All values are relative to the 1990-1999 mean. Values are shown for two regional marine ecosystem models in southeast Australia: a trophodynamic model (Ecopath with Ecosim) and a composite (hybrid) model (Atlantis). Spatial extent of models is overlapping but non-identical. Output only shown for IPSL RCP 2.6 and IPSL RCP 8.5 model runs without fishing imposed. C) Global spatial model output runs (models as per panel A) for IPSL RCP 8.5 model with fishing imposed. Ensemble model mean percentage change in total consumer carbon biomass density (g m$^{-2}$) from the 1990s to the 2050s; a positive value indicates an increase over time. Percentage changes in each grid cell for the three models in panel (A) over this time-period were averaged. See Section 3.2 for details of fisheries forcing.

**Fig. 1.**

---

## Author Response (AR2)

Dear Dr. Yool,

We are pleased to hear that the manuscript is suitable for publication, pending a minor correction. We have adjusted the colour scheme in Figure 1 as requested, to more clearly delineate areas of increasing and decreasing total consumer biomass. In addition, we have taken the opportunity to make two more minor changes for consistency. Firstly, we have also changed this map panel (Fig 1C) from an ensemble mean with fishing imposed to one without fishing imposed (just climate effects), to make consistent with panels A and B. Furthermore, we have reversed the colour scheme on Figure S2, so that a decline in NPP is coloured red (rather than an increase); thus what are typically/generally considered 'worsening' conditions (increasing SST, decreasing NPP, decreasing total consumer biomass) are always coloured red.

Sincerely,

Derek Tittensor, on behalf of all co-authors.